# RD-Suite: A Benchmark for Ranking Distillation

**Zhen Qin, Rolf Jagerman, Rama Pasumarthi, Honglei Zhuang, He Zhang, Aijun Bai, Kai Hui, Le Yan, Xuanhui Wang**[*]
Google
Mountain View, CA, US 94043

## Abstract

The distillation of ranking models has become an important topic in both academia and industry. In recent years, several advanced methods have been proposed to tackle this problem, often leveraging *ranking* information from teacher rankers that is absent in traditional classification settings. To date, there is no well-established consensus on how to evaluate this class of models. Moreover, inconsistent benchmarking on a wide range of tasks and datasets make it difficult to assess or invigorate advances in this field. This paper first examines representative prior arts on ranking distillation, and raises three questions to be answered around methodology and reproducibility. To that end, we propose a systematic and unified benchmark, Ranking Distillation Suite (RD-Suite), which is a suite of tasks with 4 large real-world datasets, encompassing two major modalities (textual and numeric) and two applications (standard distillation and distillation transfer). RD-Suite consists of benchmark results that challenge some of the common wisdom in the field, and the release of datasets with teacher scores and evaluation scripts for future research. RD-Suite paves the way towards better understanding of ranking distillation, facilities more research in this direction, and presents new challenges.

## 1 Introduction

Ranking models have become the interface between users and many applications, such as search engines and recommender systems [43, 11, 17]. In recent years, due to the popularity of large neural ranking models [22, 18, 45], as well as the interest in model serving in practical scenarios, such as on mobile devices [37], the intersection of learning to rank (LTR) [19] and knowledge distillation [10, 12] has drawn much attention in both academia and industry [32, 35, 39, 40, 44].

Examining the evaluation and experimental setup of many of these papers, we found that there is no unified consensus on what makes an acceptable test bed for benchmarking ranking distillation methods. There is also a large diversity in the types of tasks and datasets adopted, as well as the configurations of teacher and student rankers. These make comparison of different methods as well as an assessment of their relative strengths and weaknesses difficult.

In this paper, we propose a new benchmark, Ranking Distillation Suite (RD-Suite), for the purpose of benchmarking ranking distillation methods. We design a benchmark suite comprised of two major modalities in the ranking literature, sensible teacher and student model architectures, and both standard distillation and a novel distillation transfer setting. The datasets and evaluation scripts are publicly available[2] and the experimental configurations are made clear to encourage fair comparisons for future research on the important topic.

---

[*]Correspond to Zhen Qin <zhenqin@google.com>.

[2]`https://github.com/tensorflow/ranking/tree/master/tensorflow_ranking/datasets/rd_suite`

37th Conference on Neural Information Processing Systems (NeurIPS 2023) Track on Datasets and Benchmarks.

While the focus of this benchmark is an empirical comparison, we are also fundamentally interested in understanding the open questions and limitations of existing methods on ranking distillation. By examining the general formulation and prior art, we raise three questions around methodology and reproducibility, and provide discussions that challenge some of the common wisdom in the field, shine light on the empirical results, rationalize our design choices, and open up research challenges to tackle. We believe such a side-by-side performance benchmark will be valuable to the community, providing deeper insight on the practical effectiveness of different methods. The contributions and resulted artifacts of this work include:

- A systematic ranking distillation benchmark covering a variety of datasets, modalities, and methods, based on design choices around generality, simplicity, and reproducibility.
- The raise and analysis of key issues that challenge prior art and pave the way towards better understanding of ranking distillation.
- The release of easily accessible datasets and evaluation scripts to facilitate future research.

## 2 Preliminaries

We describe the general formulation of ranking distillation and representative existing arts.

### 2.1 The General Formulation

For *regular LTR*, the training data can be represented as a set $\Psi = \{(\mathbf{x}, \mathbf{y}) \in \chi^n \times \mathcal{R}^n)\}$, where $\mathbf{x}$ is a list of $n$ items $x_i \in \chi$ and $\mathbf{y}$ is a list of $n$ relevance labels $y_i \in \mathcal{R}$ for $1 \leq i \leq n$. We use $\chi$ as the universe of all items. The objective is to learn a function that produces an ordering of items in $\mathbf{x}$ so that the utility of the ordered list is maximized. Most LTR algorithms formulate the problem as learning a ranking function to score and sort the items in a list. As such, the goal of LTR boils down to finding a parameterized ranking function $f(\cdot; \theta) : \chi^n \to \mathcal{R}^n$, where $\theta$ denotes the set of trainable parameters, to minimize the empirical loss:

$$\mathcal{L}(f(\cdot; \theta)) = \frac{1}{|\Psi|} \sum_{(\mathbf{x}, \mathbf{y}) \in \Psi} l_{rel}(\mathbf{y}, f(\mathbf{x}; \theta)), \tag{1}$$

where $l_{rel}(\cdot)$ is the loss function on a single list with relevance labels.

In tabular LTR [19, 23, 30], each $x_i$ corresponds to a query-item pair represented as a numeric feature vector, and $\theta$ is usually a linear, tree, or multilayer perception (MLP) model. For text ranking [22, 45], each $x_i$ represents query and document text strings and transformer-based models, such as BERT [8] and T5 [31], are the norm for $\theta$.

For *ranking distillation*, in addition to the original training data, it is assumed that there is a teacher ranker $f(\cdot; \theta^t)$ producing teacher scores (or distillation labels) $\mathbf{y}^t = g(f(\cdot; \theta^t))$, where $g(\cdot)$ is an optional (ranking preserving) transform function on teacher scores. The goal of ranking distillation is to train a student model $f(\cdot; \theta^s)$, where $\theta^s$ usually has significantly smaller capacity than $\theta^t$, with the following loss:

$$\mathcal{L}(f(\cdot; \theta^s)) = \frac{1}{|\Psi|} \sum_{(\mathbf{x}, \mathbf{y}) \in \Psi} \left( \alpha \times l_{rel}(\mathbf{y}, f(\mathbf{x}; \theta^s)) + (1 - \alpha) \times l_{distill}(\mathbf{y}^t, f(\mathbf{x}; \theta^s)) \right), \tag{2}$$

where $\alpha \in [0, 1]$ is a weighting factor and $l_{distill}$ is the additional distillation loss.

### 2.2 Existing Art

Regular LTR research has proposed a rich set of $l_{rel}(\cdot)$, including pointwise, pairwise, and listwise losses [19]. Here we describe two recent most representative ranking distillation work (see more discussions in Sec. 7) that focus on $l_{distill}$.

**RD [35]** treats top $K$ ranked items by the teacher ranker as positive examples (no negatives are considered), and apply a *pointwise* sigmoid cross-entropy loss:

$$l_{\mathrm{RD}}(\mathbf{y}^t, f(\mathbf{x}; \theta^s)) = -\sum_{i=1}^{K} \log f(x_{\pi(i)}; \theta^s), \tag{3}$$

where $\pi(i)$ is the $i$-th document in the ordering $\pi$ induced by teacher ranker.

**RankDistil [32]** can be treated as a listwise generalization of RD, trying to preserve *ordering* of the top $K$ items induced by the teacher ranker:

$$l_{\text{RankDistil}}(\mathbf{y}^t, f(\mathbf{x}; \theta^s)) = E_{\pi \sim P_{\mathbf{y}^t}}[-\log P_s(\pi, f(\mathbf{x}; \theta^s))], \tag{4}$$

where

$$P_s(\pi, f(\mathbf{x}; \theta^s)) = \frac{1}{(L-K)!} \prod_{j=1}^{K} \frac{\exp(f(x_{\pi(j)}; \theta^s))}{\sum_{l=j}^{L} \exp(f(x_{\pi(l)}; \theta^s))} \tag{5}$$

is the Plackett-Luce probability model, $L$ is the length of the ranking list, and $\pi(j)$ is the $j$-th document from a *sampled* ordering $\pi$. Essentially, it samples several length-$K$ permutations from teacher ranker score distribution ($P_{\mathbf{y}^t}$) and tries to maximize the log likelihood of the induced permutations from student scores of the corresponding documents.

# 3 Three Questions on Ranking Distillation

By examining the general formulation and prior art, we raise three questions about the current literature on ranking distillation.

## 3.1 Q1: Where is the knowledge in ranking distillation?

An implicit assumption of RD, RankDistil, and related methods is, the knowledge for distillation [12] is in the (pointwise or listwise) top $K$ *orderings*.

There are three concerns of this assumption: First, they largely ignore the teacher score *values*[3]. In contrast, in classification distillation, the logit values of all classes are considered [12]. Importantly, ordering can be derived from score values, but not the other way. Is it possible that some knowledge is lost by only considering the ordering? Second, the top $K$ is a hyperparameter. Besides more tuning efforts, it is counter-intuitive that it should be the same for all lists in a dataset, due to the potentially huge variance in queries and their corresponding documents in practice. Third, documents outside of top $K$ may be useful and contribute to the knowledge. The fact that prior work mainly compared under this assumption (e.g., RankDistil mainly compared with RD) makes the answer to the questions unclear.

## 3.2 Q2: How to handle teacher ranker scores?

As we argued to potentially leverage teacher score values above, the teacher scores need closer examination. In general, the teacher ranker scores may not be constrained – the ranks of candidate documents are invariant to any transformation that preserves the order of the scores. An example is the scale invariance of many advanced loss functions:

- The pairwise logistic loss in RankNet [4]:

$$l_{\text{PairLog}}(\mathbf{y}, \mathbf{s}) = -\sum_{i \neq j} \mathbb{I}_{y_i > y_j} \ln \frac{\exp(s_i - s_j)}{1 + \exp(s_i - s_j)}. \tag{6}$$

- The listwise Softmax loss [5, 3]:

$$l_{\text{Softmax}}(\mathbf{y}, \mathbf{s}) = -\sum_i y_i \ln \frac{\exp(s_i)}{\sum_j \exp(s_j)} = -\sum_i y_i \ln \frac{1}{\sum_j \exp(s_j - s_i)},$$

where we use $\mathbf{s} = f(\mathbf{x}; \theta)$ for conciseness. In these popular ranking losses, scores always appear in the *paired* form of $s_i - s_j$ or $s_j - s_i$. So when a constant is added to each score, the losses stay invariant. Many other ranking losses, such as Approximate NDCG loss [27, 2], also have the same translation-invariant property.

---

[3]Strictly speaking, ordering depends on the values. In this work we differentiate between absolute values and ordering.

This is in contrast to classification problems, where the logits have clear probabilistic meanings over all classes. The potentially unconstrained teacher scores need careful treatment. For example, many loss functions (such as cross-entropy based ones) are ill-defined with negative labels, potentially leading to devastating downstream distillation performance.

### 3.3 Q3: How to ensure reproducible and fair evaluations?

We note several complexities to ensure reproducible and fair evaluation for ranking distillation research. First, for datasets with large corpus, a retrieval stage is performed before ranking, which changes the data distribution for the ranking stage. Second, it is clear that the teacher model has significant effect on the downstream distillation task. For example, as we discussed, teacher ranker scores have large freedom and are difficult to reproduce. Third, researchers should be careful about what to ablate to focus on progress on distillation methods. In Eq. 2, we can see that (ranking) distillation is multi-objective. Though classification problems almost universally use the softmax cross-entropy loss for $l_{\text{rel}}$, ranking problems have more flexibility for this term. In [32], the two loss terms always change together - it is unclear if the performance differences are from a better relevance loss or distillation method. Last but not least, there is no consensus on how to calculate ranking metrics. For example, queries without any relevant documents may get perfect score [25], 0 [13], or ignored [23]. Different metric definition makes cross-examination among papers difficult.

## 4 RD-Suite

We describe RD-Suite, starting with a set of desiderata motivated by [36], followed by the actions we take, and how they link to the desiderata and questions raised in the previous section.

### 4.1 Desiderata

For creating the RD-Suite, we established a set of desiderata:

D1. Generality: Given the long history of learning to rank research on tabular datasets, as well as the more recent interest in text ranking, RD-Suite includes tasks in both modalities. Also, we focus on distillation objectives in this paper, instead of model architecture specific distillation techniques (see Sec. 7), though the latter should benefit from better objective functions and easily compare against the benchmark with appropriate ablations.

D2. Simplicity: The tasks should have a simple setup. All factors that make comparisons difficult should be removed. This encourages simple models instead of cumbersome pipelined approaches. For instance, we consider pretraining to be out of scope of this benchmark. The student rankers are either simple to implement or initiated from publicly available checkpoints.

D3. Challenging: The tasks should be difficult enough for current models to ensure that there is room for improvement to encourage future research in this direction. We leverage state-of-the-art teacher rankers that still have good advantage over existing methods. We also introduce a novel distillation transfer task with encouraging results but considerable headroom.

D4. Non-resource intensive and accessible: The benchmarks should be deliberately designed to be lightweight so as to be accessible to researchers without industry-grade computing resources. We consider linear students for tabular ranking and regular public BERT models for text ranking. We handle the more expensive teacher ranker training and inference, and directly expose the distillation dataset to the community.

D5. Fairness. We note that it is non-trivial and almost impossible to conduct a perfectly fair evaluation of all models. The large search space motivates us to follow a set of fixed hyperparameters for most models (discussed in Sec. 4.6 with few exceptions). The best performance and relative order of the models may change if we aggressively tune hyperparameters for all models. Hence, the results provided in this paper are not meant to be a final authoritative document on which method is the best. Instead, we provide a starting point for future research and strive to be as fair as possible with the tuning process clearly laid out.

## 4.2 Tasks

RD-Suite has four tasks, differing in terms of data modality, teacher model domain, and availability of relevance label. They are described in Table 1. Specifically, T1 (Text Ranking Distillation) and T4 (Tabular Ranking Distillation) are standard ranking distillation tasks on two common modalities in the LTR literature. T2 (Distillation Transfer) and T3 (Distillation Transfer Zeroshot) are motivated by the popularity of domain adaptation research where knowledge is transferred from source to target domains [31, 18]. We believe the tasks are comprehensive and reflect real-world applications: T1 and T4 are standard ranking distillation tasks, T2 is applicable when target domain data is not sufficient to tune the teacher model, or the teacher model is not accessible for tuning. T3 is applicable when there are no labels for the target domain, which is the zeroshot learning scenario [38]. Note that domain adaptation is not applicable to tabular LTR since different datasets are in different feature spaces.

Table 1: The tasks of RD-Suite. In-domain means teacher was trained on the same dataset as the student and out-domain otherwise.

| Tasks | Modality | Teacher Domain | Relevance Label |
|-------|----------|----------------|-----------------|
| T1 | Text | In-domain | Available |
| T2 | Text | Out-domain | Available |
| T3 | Text | Out-domain | Not Available |
| T4 | Tabular | In-domain | Available |

The tasks are designed to cover wide application scenarios (D1) while maintaining simplicity (D2) - for example, T1, T2, and T3 can use the same implementation by changing teacher label and $\alpha$ (T3 simply has $\alpha = 0$). The novel ranking distillation transfer setting is quite challenging (D3) and not well studied in the literature.

## 4.3 Teacher and Student Configurations

We make the teacher and student ranker configurations clear. For text datasets, the teacher rankers are from RankT5 [45], which use encoder-decoder T5 [31] and listwise softmax ranking losses. For tabular datasets, the teacher rankers are from [13], which use MLP and a novel LambdaLoss that will discussed below. To our knowledge, they are arguably the state-of-the-art in their respective tasks and we get the teacher models from the authors. As for student rankers, we use the publicly available BERT-base [8] checkpoint for text datasets and linear model for tabular datasets.

The advanced teacher rankers provide reasonable headroom (D3) and the student configurations follow D2 and D5 - they are easily accessible or easy to implement, and can be trained with accessible computing resources.

## 4.4 Datasets

All datasets in RD-Suite are popular public datasets. For text ranking, we use MSMARCO [1] and NQ [16], two of the most popular datasets for text ranking. Both datasets have large document corpus and require a retrieval stage before ranking. We leverage recent neural retrievers [20] to retrieve the top 50 candidates for each query.

For tabular ranking, we use Web30K [26] and Istella [7], two of the most popular datasets for tabular ranking. Retrieval is not needed for these datasets.

RD-Suite will release the teacher scores and retrieved document ids (on MSMARCO and NQ) for each query, to hide the potentially expensive teacher model inference and retrieval from ranking distillation research (D4) while guaranteeing fairness (D5, Q3).

## 4.5 The Ranking Loss Family for Distillation

As we discussed in Q1 (Where is the knowledge in ranking distillation?), there are several concerns around existing ranking distillation methods, namely they do not consider score values, has a top $K$ hyperparameter, and largely ignore documents outside of the top $K$.

Interestingly, we find that many *ranking* losses naturally have different behaviors around the concerns, but are not well studied in the distillation context. Therefore, we include the "ranking loss family"

for distillation in RD-Suite to study their effects and gain better understanding of assumptions of existing methods. We include six ranking losses that are some of the most representative methods in the literature, including the pointwise mean squared error (MSE), pairwise logistic loss (PairLog in Eq. 6) and mean squad error (PairMSE):

$$l_{\text{PairMSE}}(\mathbf{y}, \mathbf{s}) = \sum_{i \neq j} ((s_i - s_j) - (y_i - y_j))^2, \tag{7}$$

as well as the listwise Softmax (Eq. 7), Gumbel Approximate NDCG loss (GumbelNDCG) [2]:

$$l_{\text{GumbelNDCG}}(\mathbf{y}, \mathbf{s}) = -\frac{1}{\widetilde{\text{DCG}}} \sum_i \frac{2^{y_i} - 1}{\log_2(1 + r_i)}, \tag{8}$$

where $\widetilde{DCG}$ is the ideal DCG metric, $r_i$ is the approximate rank of document $i$ given $\mathbf{s}$ plus Gumbel noise, and LambdaLoss [13]:

$$l_{\text{LambdaLoss}}(\mathbf{y}, \mathbf{s}) = -\sum_{i \neq j} \mathbb{I}_{y_i > y_j} \Delta_{ij} \ln \frac{\exp(s_i - s_j)}{1 + \exp(s_i - s_j)}, \tag{9}$$

where $\Delta_{ij}$ is the LambdaWeight as defined in Eq.11 of [13].

## 4.6 Model Tuning

We fix $l_{\text{rel}}$ for all methods on each modality to focus on the distillation component (Q3). To study Q2, we find that the Softmax transformation is naturally order preserving and ensures positive labels, motivated by the classification setting [12][4] and the RankDistil work [32] :

$$g(f(\mathbf{x}; \theta^t)) = \frac{\exp(f(\mathbf{x}; \theta^t)/T)}{\sum_i \exp(f(x_i; \theta^t)/T)}, \tag{10}$$

where $T$ is the temperature. However, we note that some losses do not require positive labels (such as MSE), so we have the option to not use it (see more discussion in Sec. 5.5). We list the hyperparameter search space in Tbl. 2.

Table 2: Hyperparameter search space.

| Data | Text | Tabular |
|---|---|---|
| $l_{rel}$ | Softmax | LambdaLoss |
| Optimizer | AdamW | Adagrad |
| Learning Rate | 1e-4, 1e-5, 1e-6 | 1e-1, 1, 10 |
| Batch Size (of Lists) | 32 | 128 |
| Training Steps | 100K | 200K |
| Sequence Length | 128 | n/a |
| Head Weight $\alpha$ | 0, 0.25, 0.5, 0.75, 1 | |
| Label Softmax Transform | on, off | |
| Temperature T | 0.1, 1.0, 2.0, 5.0, 10.0 | |

AdamW is the default optimizer for BERT. The key principle of parameter search space is, all methods share the same tuning budget (D5). The only exception is RD and RankDistill: since they have an additional hyperparameter $K$, we sweep $K \in \{1, 5, 10\}$ and thus they have *advantage* in terms of tuning budgets.

All methods are implemented in the same open-source framework TF-Ranking [24]. All methods in the ranking loss family have already been implemented in TF-Ranking. We implemented RD due to its simplicity. We got the RankDistil implementation from the authors and had several communications to make sure of its correctness. Using open-source implementations encourage simplicity and fairness (D2, D5).

---

[4]Softmax transformation is over the list of classes for each item in classification and over the list of items in ranking.

Table 3: Ranking distillation results on the MSMARCO and NQ datasets. $\uparrow$ means significantly better result, performed against "Relevance Only" at the $p < 0.01$ level using a two-tailed $t$-test. Best model is in boldface and second best is underlined for each metric (except for the teacher).

| Models | MSMARCO | | | | | NQ | | | | |
|---|---|---|---|---|---|---|---|---|---|---|
| | MRR@10 | MRR | NDCG@1 | NDCG@5 | NDCG | MRR@10 | MRR | NDCG@1 | NDCG@5 | NDCG |
| Teacher | 43.63 | 44.46 | 29.91 | 46.84 | 54.22 | 60.08 | 60.36 | 46.95 | 63.89 | 66.98 |
| Relevance Only | 40.03 | 41.03 | 27.15 | 42.85 | 51.29 | 52.67 | 53.50 | 43.47 | 55.07 | 60.98 |
| RD | 40.25 | 41.25 | 27.51 | 42.92 | 51.45 | 57.09$^\uparrow$ | 57.64$^\uparrow$ | 46.72$^\uparrow$ | 59.80$^\uparrow$ | 64.50$^\uparrow$ |
| RankDistil | 41.29$^\uparrow$ | 42.21$^\uparrow$ | 28.27$^\uparrow$ | 44.04$^\uparrow$ | 52.28$^\uparrow$ | 55.04$^\uparrow$ | 55.80$^\uparrow$ | 45.71$^\uparrow$ | 57.53$^\uparrow$ | 62.86$^\uparrow$ |
| MSE | 42.06$^\uparrow$ | 42.96$^\uparrow$ | 28.54$^\uparrow$ | 45.12$^\uparrow$ | 52.96$^\uparrow$ | 58.49 | 58.95$^\uparrow$ | 47.51$^\uparrow$ | 61.43$^\uparrow$ | 65.64$^\uparrow$ |
| PairLog | 41.95$^\uparrow$ | 42.87$^\uparrow$ | 28.21 | 45.00$^\uparrow$ | 52.86$^\uparrow$ | 58.53$^\uparrow$ | 59.00$^\uparrow$ | 47.95$^\uparrow$ | 61.38$^\uparrow$ | 65.66$^\uparrow$ |
| PairMSE | 42.27$^\uparrow$ | 43.20$^\uparrow$ | 28.78$^\uparrow$ | 45.19$^\uparrow$ | 53.11$^\uparrow$ | 58.34$^\uparrow$ | 58.79$^\uparrow$ | 47.05$^\uparrow$ | 61.44$^\uparrow$ | 65.55$^\uparrow$ |
| GumbelNDCG | 41.85$^\uparrow$ | 42.76$^\uparrow$ | 28.32$^\uparrow$ | 44.81$^\uparrow$ | 52.75$^\uparrow$ | 57.90$^\uparrow$ | 58.39$^\uparrow$ | 47.26$^\uparrow$ | 60.70$^\uparrow$ | 65.14$^\uparrow$ |
| Softmax | **42.37**$^\uparrow$ | **43.27**$^\uparrow$ | **28.87**$^\uparrow$ | 45.33$^\uparrow$ | **53.17**$^\uparrow$ | **59.08**$^\uparrow$ | **59.58**$^\uparrow$ | **48.84**$^\uparrow$ | **61.87**$^\uparrow$ | **66.08**$^\uparrow$ |
| LambdaLoss | 42.30$^\uparrow$ | 43.21$^\uparrow$ | 28.75$^\uparrow$ | **45.38**$^\uparrow$ | 53.14$^\uparrow$ | 58.36$^\uparrow$ | 58.85$^\uparrow$ | 47.87$^\uparrow$ | 61.13$^\uparrow$ | 65.51$^\uparrow$ |

# 5 Results

We report results and observations on the four tasks in RD-Suite. We use some of the most popular ranking metrics, i.e., NDCG and MRR, on the relevance labels. The numbers are timed by 100 which is common in the literature. The graded labels in Web30K and Istella are binarized at 3 when computing MRR that requires binary labels. We first discuss the results on each task specifically and summarize more findings on all tasks in Sec. 5.4. All evaluation scripts are made available to facilitate cross paper comparisons (Q3).

## 5.1 Results on Text Ranking Distillation (T1)

Tbl. 3 shows the ranking performance on MSMARCO and NQ datasets. We have the following observations: (1) The effectiveness of existing methods (RD and RankDistil) is verified. RankDistil significantly outperforms the Relevance Only baseline in both datasets while RD is effective on NQ. (2) However, the performance of RD and RankDistil are considerably worse compared with the ranking loss family. (3) The listwise Softmax loss tends to be the most robust method. It is encouraging to see some student models can even outperform the teacher on NQ NDCG@1.

## 5.2 Results on Distillation Transfer (T2 and T3)

Tbl. 4 shows the ranking performance on the NQ dataset when transferring knowledge from the teacher rankers trained on MSMARCO (the same MSMARCO teacher in T1). For NQ-Zeroshot, since the relevance labels are not available, the Relevance Only baseline is not applicable. RD is also not applicable since it has to use relevance labels (all labels are simply positive in the distillation objective).

We have the following observations: (1) Results on the distillation transfer task are interesting and encouraging: even if the teacher ranker does not perform well, all distillation methods can get significant help from it and outperform the Relevance Only baseline. Note that the "poor teacher helps distillation" effect has drawn some attention in the classification setting [42] but is not well understood for ranking problems. (2) Results on NQ-Zeroshot are expected - as no relevance labels are available for the target domain, the teacher model becomes the headroom. There is still considerate performance variance among different methods, showcasing their capabilities in a "distillation only" setting. (3) Existing methods (RD and RankDistil) are still not competitive. (4) Listwise approaches such as Softmax and GumbelNDCG tend to be effective.

## 5.3 Results on Tabular Ranking Distillation (T4)

Tbl. 5 shows the ranking performance on the Web30K and Istella datasets. Despite data modality and model configurations, one major difference is tabular LTR datasets have dense relevance labels – multiple documents with non-zero relevance, while for MSMARCO and NQ there is usually one labelled positive document for each query.

Table 4: Distillation transfer results on the NQ dataset, using Teacher model fine-tuned on MS-MARCO. NQ-Zeroshot means relevance label on NQ is not available. $\uparrow$ means significantly better result, performed against "Relevance Only" at the $p < 0.01$ level using a two-tailed $t$-test. Best model is in boldface and second best is underlined for each metric (except for the teacher).

| Models | NQ | | | | | NQ-Zeroshot | | | | |
|---|---|---|---|---|---|---|---|---|---|---|
| | MRR@10 | MRR | NDCG@1 | NDCG@5 | NDCG | MRR@10 | MRR | NDCG@1 | NDCG@5 | NDCG |
| Teacher | 45.27 | 46.02 | 31.08 | 48.94 | 55.49 | 45.27 | 46.02 | 31.08 | 48.94 | 55.49 |
| Relevance Only | 52.67 | 53.50 | 43.47 | 55.07 | 60.98 | n/a | n/a | n/a | n/a | n/a |
| RD | 54.66$^\uparrow$ | 55.25$^\uparrow$ | 44.25 | 57.34$^\uparrow$ | 62.57$^\uparrow$ | n/a | n/a | n/a | n/a | n/a |
| RankDistil | 54.17$^\uparrow$ | 54.95$^\uparrow$ | 44.19 | 56.88$^\uparrow$ | 62.28$^\uparrow$ | 40.01 | 41.10 | 27.67 | 42.95 | 51.25 |
| MSE | 56.03$^\uparrow$ | 56.64$^\uparrow$ | 46.12$^\uparrow$ | 58.66$^\uparrow$ | 63.64$^\uparrow$ | 43.20 | 44.09 | 29.49 | 46.76 | 53.84 |
| PairLog | 56.23$^\uparrow$ | 56.88$^\uparrow$ | 46.58$^\uparrow$ | 58.76$^\uparrow$ | 63.79$^\uparrow$ | 43.37 | 44.25 | 29.72 | 46.84 | 53.94 |
| PairMSE | 56.25$^\uparrow$ | 56.89$^\uparrow$ | 46.58$^\uparrow$ | 58.91$^\uparrow$ | 63.84$^\uparrow$ | 43.45 | 44.35 | 29.82 | 47.05 | 54.05 |
| GumbelNDCG | 56.03$^\uparrow$ | 56.70$^\uparrow$ | 46.43$^\uparrow$ | 58.72$^\uparrow$ | 63.62$^\uparrow$ | **44.00** | **44.87** | **30.85** | **47.38** | **54.39** |
| Softmax | **56.72**$^\uparrow$ | **57.33**$^\uparrow$ | **47.09**$^\uparrow$ | **59.30**$^\uparrow$ | **64.16**$^\uparrow$ | 43.80 | 44.66 | 30.13 | 47.28 | 54.27 |
| LambdaLoss | 56.41$^\uparrow$ | 57.06$^\uparrow$ | 46.98$^\uparrow$ | 58.80$^\uparrow$ | 63.93$^\uparrow$ | 43.78 | 44.61 | 30.22 | 47.26 | 54.23 |

We have the following observations: (1) There are generally fewer effective distillation methods on these tabular datasets than the text datasets, especially on Web30K. We hypothesize it is because of the dense relevance labels, so the benefits from teacher scores are less clear. (2) Listwise ranking losses such as Softmax and LambdaLoss are the most competitive.

Table 5: Ranking distillation results on the Web30K and Istella datasets. $\uparrow$ means significantly better result, performed against "Relevance Only" at the $p < 0.01$ level using a two-tailed $t$-test. Best model is in boldface and second best is underlined for each metric (except for the teacher).

| Models | Web30K | | | | | Istella | | | | |
|---|---|---|---|---|---|---|---|---|---|---|
| | MRR@10 | MRR | NDCG@1 | NDCG@5 | NDCG | MRR@10 | MRR | NDCG@1 | NDCG@5 | NDCG |
| Teacher | 34.43 | 35.05 | 48.39 | 47.33 | 71.62 | 84.54 | 84.59 | 71.69 | 67.92 | 81.69 |
| Relevance Only | 27.81 | 28.54 | 41.43 | 41.07 | **68.37** | 76.97 | 77.12 | 61.09 | 57.18 | 74.23 |
| RD | 27.44 | 28.19 | 41.17 | 40.92 | 68.32 | 77.01 | 77.16 | 61.36 | 57.35$^\uparrow$ | 74.32$^\uparrow$ |
| RankDistil | 27.82 | 28.58 | 41.47 | 41.03 | 68.33 | 77.15 | 77.30 | 61.61$^\uparrow$ | 57.41$^\uparrow$ | 74.36$^\uparrow$ |
| MSE | 27.44 | 28.19 | 41.37 | 40.92 | 68.31 | 77.31$^\uparrow$ | 77.46$^\uparrow$ | 61.89$^\uparrow$ | 57.52$^\uparrow$ | **74.41**$^\uparrow$ |
| PairLog | 27.51 | 28.25 | 41.10 | 40.98 | 68.28 | 77.04 | 77.19 | 61.34 | 57.37$^\uparrow$ | 74.32$^\uparrow$ |
| PairMSE | 26.97 | 27.72 | 40.70 | 40.90 | 68.24 | 77.17 | 77.32 | 61.66$^\uparrow$ | 57.51$^\uparrow$ | 74.37$^\uparrow$ |
| GumbelNDCG | 28.31$^\uparrow$ | 29.05$^\uparrow$ | 41.79 | **41.29** | 68.32 | 77.44$^\uparrow$ | 77.63$^\uparrow$ | **62.96**$^\uparrow$ | 57.53$^\uparrow$ | 74.06 |
| Softmax | **29.54**$^\uparrow$ | **30.25**$^\uparrow$ | **42.08** | 41.11 | 68.22 | 77.49$^\uparrow$ | 77.64$^\uparrow$ | 62.47$^\uparrow$ | 57.69$^\uparrow$ | 74.40$^\uparrow$ |
| LambdaLoss | 29.31$^\uparrow$ | 30.02$^\uparrow$ | 42.03 | 41.13 | 68.30 | **77.59**$^\uparrow$ | **77.75**$^\uparrow$ | 62.56$^\uparrow$ | **57.75**$^\uparrow$ | **74.41**$^\uparrow$ |

## 5.4 Overall Results

To get an overview of distillation methods, in Tbl. 6, we calculate their performance ranks on the 6 configurations, based on the NDCG@5 metric.

Table 6: The performance ranks of different distillation methods on the 6 task configurations. Smaller rank indicates better performance.

| Model | Best Rank | Worst Rank | Mean Rank |
|---|---|---|---|
| Softmax | 1 | 3 | 1.8 |
| LambdaLoss | 1 | 5 | 2.7 |
| GumbelNDCG | 1 | 6 | 3.7 |
| PairMSE | 1 | 8 | 3.8 |
| MSE | 3 | 6 | 4.8 |
| PairLog | 4 | 7 | 5.0 |
| RankDistil | 4 | 8 | 6.7 |
| RD | 6 | 8 | 7.2 |

There are several trends: first, listwise ranking losses (Softmax, LambdaLoss, GumbelNDCG) that consider the entire list of documents and teacher score values are most competitive. PairMSE and MSE being next shows the importance of teacher score values over PairLog. Existing methods (RD and RankDistil) are not competitive despite they are effective distillation methods in general (outperforming no distillation baseline in most cases.)

## 5.5 The Role of Softmax Transformation

To better understand Q2 (How to handle teacher ranker scores?), in experiments we have a switch on whether to apply softmax transformation and report best performing configurations in the results. Here we peek into if softmax label transformation is enabled for each method.

We examine the MSMARCO and NQ ranking distillation experiments (T1) and observe the same behavior: softmax label transformation was selected by GumbelNDCG, Softmax, and LambdaLoss, but was *not* selected for MSE and PairMSE. Note that PairLog and RD are indifferent to such order preserving transformation since they do not look at teacher score values, and Rankdistill has to use softmax transformation to sample distributions as done in the original paper.

Some observations are intuitive: cross-entropy based losses such as Softmax are ill-behaved with negative labels; GumbelNDCG and LambdaLoss consist of DCG computation that usually assumes positive labels. The behaviors of MSE and PairMSE are interesting - it indicates methods depending heavily on score values can be less effective with certain transformations. In fact, the performance gap is quite significant: the MRR@10 of MSE on NQ is 54.87 vs 58.49 with softmax transformation turned on and off, a difference between being highly competitive (3rd best result for that task) and not so. We provide more discussions in Appendix. Therefore, researchers should be cautious about label transformations on teacher ranker scores that have much more freedom that those in classification problems, and we provide more discussions in Sec. 6.

## 5.6 The Learning Dynamics

We draw the ranking performance on validation data during model training to gain insight in the learning dynamics in Fig. 1. We can see that when compared against relevance labels that we care most, the Relevance Only baseline saturates fast, while distillation denoted as Softmax (using Softmax as the distill loss) performance keeps increasing with the help of distillation. Distillation can also get good ranking performance against teacher scores, but it is interesting that it does not have to keep increasing performance on teacher scores to increase performance on relevance labels. Relevance Only model's behavior on teacher scores is also interesting: the performance can even get worse during training (note the Relevance Only model does not have access to teacher scores). Our understanding is this indicates some overfitting issues that can not be caught by relevance labels.

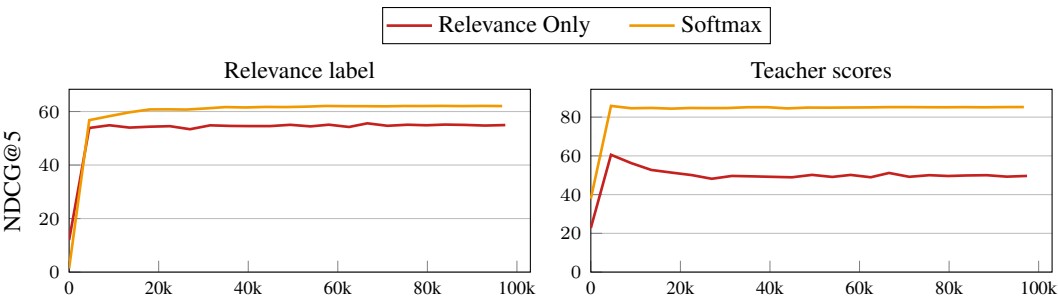

Figure 1: The performance of Relevance Only and Softmax on relevance label and teacher scores for the NQ ranking distillation task. NDCG is used since teacher scores are real-valued.

## 6 Summary and Discussions

We provide a brief summary that answers the three questions we raised in Sec. 3. We then discuss topics that could be important for ranking distillation, but are not comprehensively covered in the current benchmark due to scope of this work.

**Summary**. For Q1: Where is the knowledge in ranking distillation, we performed comprehensive benchmark and found that the knowledge can exist in the score values from teacher rankers, challenging common assumptions made in the literature. We hope this can inspire future work in this direction. For Q2: How to handle teacher ranker scores, we showed the freedom of practical teacher rankers that is not well studied, and the effect of label transformations that were either ignored or implicitly considered in the literature. This problem also needs more future work considering the

importance of teacher score values in Q1. For Q3, we discussed drawbacks of existing efforts, and strove to provide a standardized criterion, by providing retrieved documents, teacher scores, student architectures, and the evaluation scripts.

**The role of teacher ranker**. We assume teacher rankers are given, which is the common setting in the literature, motivating us to release the teacher scores. However, it is still meaningful to investigate different teacher rankers and their effect in the distillation process. Some teacher ranker may not be state-of-the-art itself but may produce very competitive student ranker (as hinted in the distillation transfer experiments). Recently, there are studies around this in the classification setting [21].

**The role of teacher label transformation**. We formalized the importance of properly handling teacher ranker labels. We studied Softmax transformation motivated by traditional knowledge distillation and some existing work, but also argued that such transformation may not always be useful. An open question is if there exist better teacher label transformations.

**What tasks need more attention?** RD-Suite consists of a diverse set of tasks. However, we encourage researchers to focus more on the text ranking tasks (T1, T2, T3) for the following reasons: (1) The setting on tabular datasets can be useful from a research perspective but less from a practical perspective - MLP models may not be expensive to serve, and the community has yet to find ways to train large effective models on tabular datasets [29]. Other models such as Gradient Boosted Decision Trees (GBDTs) are also highly effective alternatives [30]. (2) Tabular datasets have dense relevance labels that may make distillation less useful. Also, having human annotated dense relevance labels is costly and not practical. (3) We can not study interesting topics such as distillation transfer since each tabular dataset has different manually designed numeric features.

On the other side, we believe the text ranking tasks are more realistic by using giant teacher models, allowing knowledge transfer, and benefiting more from distillation given sparse relevance labels. For future work, we are interested in extending RD-Suite to Large Language Model based rankers [28].

# 7 Related Work

Ranking distillation has drawn much attention in the community from different perspectives. The most relevant existing work are RD [35] and RankDistil [32] that we discussed in the paper, focusing on general distillation objectives, that can be applied to any model architectures and ranking applications.

Some work focus on architecture specific distillation or model compression. For example, [9] focused on BERT-based models and perform distillation on language model tasks (i.e., predictions on all tokens in the vocabulary). Both teacher and student rankers are BERT models, which is not very general. [41] distilled intermediate representations in BERT models, and simply assumed the teacher ranker was trained using pointwise loss and applied pointwise logistic loss for distillation. There is also a body of work on BERT specific distillation in general, not limited to ranking [33].

There is a suite of work on distillation for recommender systems that rank items for users, potentially to be served on mobile devices. These work either apply RD or RankDistil as a core component, or take a route specific to the recommendation problem. [14] was a parallel work as RankDistil and performed order preserving distillation. [15] studied student to teacher distillation while the distillation component followed RD. [34, 37] concerned about distillation of sequential recommender systems and also assumed simple teacher configurations and distillation methods (e.g., pointwise losses for both teacher and distillation). [6] mitigated popularity bias for distillation that is specific to recommender systems.

# 8 Conclusion

We proposed Ranking Distillation Suite (RD-Suite), a new benchmark for evaluating progress on ranking distillation research. Our new benchmark is challenging and probes at model capabilities in dealing with diverse data types and both regular ranking distillation and distillation transfer tasks. For the first time, we conduct an extensive side-by-side comparison of teachers, baseline without distillation, and eight distillation methods. The experimental results challenge common wisdom of prior art, show promises for certain methods, and leaves room for improvements. The benchmark also includes datasets with teacher labels and evaluation scripts to facilitate future benchmarking, research, and model development.

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
