# Appendix for RD-Suite: A Benchmark for Ranking Distillation

**Zhen Qin, Rolf Jagerman, Rama Pasumarthi, Honglei Zhuang, He Zhang, Aijun Bai,
Kai Hui, Le Yan, Xuanhui Wang**
Google
Mountain View, CA, US 94043

## 1   Datasets and Evaluation

RD-Suite releases the teacher scores and retrieved document ids (on MSMARCO and NQ) for each query, to hide the potentially expensive teacher model inference and retrieval from ranking distillation research (D4) while guaranteeing fairness (D5, Q3).

### 1.1   Link

RD-Suite is hosted at `https://github.com/tensorflow/ranking/tree/master/tensorflow_ranking/datasets/rd_suite`.

### 1.2   Dataset format

The website hosts the datasets in TREC format, which is standard in information retrieval literature. More specifically we have two files for each dataset. For trec_run.txt, each role looks like:

1101282 Q0 8007514 1 3.5860724449157715 msmarco_dev_teacher

Q0, 1, and msmarco_dev_teacher are just placeholder strings, which existing (public) TREC tools require, though they do not provide any meaningful information. The other three fields are query_id, document_id, and teacher_score.

For trec_qrel.txt, each row looks like

1101282 0 8007514 1

0 is again a placeholder meaning nothing. The last number is the label field. public TREC tools will automatically join these two files using the query and document ids.

### 1.3   Evaluation Colab

We provide an evaluation colab to produce different ranking metrics using the same criterion in the paper. The evaluation colab is implemented using open-source operation, one should be easy to run the script end-to-end by following the instructions.

### 1.4   Dataset sources

All datasets in RD-Suite are popular public datasets. For text ranking, we use MSMARCO [1] and NQ [3], two of the most popular datasets for text ranking. Both datasets have large document corpus and require a retrieval stage before ranking. We leverage recent neural retrievers [4] to retrieve the top 50 candidates for each query.

For tabular ranking, we use Web30K [5] and Istella [2], two of the most popular datasets for tabular ranking. Retrieval is not needed for these datasets. They can be easily accessed via

Submitted to the 37th Conference on Neural Information Processing Systems (NeurIPS 2023) Track on Datasets and Benchmarks. Do not distribute.

TF dataset (`https://www.tensorflow.org/datasets/catalog/mslr_web` and `https://www.tensorflow.org/datasets/catalog/istella`).

## 1.5 Teacher score distribution

We report statistics of the teacher scores for each dataset and configuration in Tbl. 1. We can see the teacher scores can have wide ranges in different configurations.

Table 1: The statistics of teacher ranker scores for each configuration used in RD-Suite. The 3rd row has Dataset as NQ and Teacher as MSMARCO, and corresponds to the distillation transfer setting. x% means percentile.

| Dataset | Teacher | Mean | Std | Min | 25% | 50% | 75% | Max |
|---------|---------|------|-----|-----|-----|-----|-----|-----|
| MSMARCO | MSMARCO | -4.07 | 5.43 | -20.09 | -8.12 | -3.41 | 0.40 | 6.36 |
| NQ | NQ | -24.74 | 22.27 | -63.08 | -43.32 | -28.84 | -9.18 | 84.36 |
| NQ | MSMARCO | -7.29 | 5.10 | -19.30 | -11.23 | -7.48 | -3.54 | 5.82 |
| Web30K | Web30K | -0.33 | 0.91 | -8.57 | -0.66 | -0.17 | 0.04 | 11.86 |
| Istella | Istella | -17.26 | 15.84 | -181.54 | -21.87 | -12.38 | -7.56 | 18.63 |

# 2 Additional Experimental Results

## 2.1 Sensitivity on the loss weight $\alpha$

We examine ranking performance sensitivity on the loss weight $\alpha$. For clarity, we pick MSE, PairMSE, Softmax (one from pointwise, pairwise, and listwise losses each), and RankDistil. We plot MRR@10 on MSMARCO and NQ ranking distillation tasks (T1) with varying $\alpha$ in Fig. 1. In general the comparative performance is robust among methods. It is interesting to see different behaviors of different methods. For example, RankDistil seems to suffer without relevance information ($\alpha = 0$) but can still help when both relevance and distillation objectives are used. In most cases, best performance is achieved when both objectives are active, except for Softmax on MSMARCO.

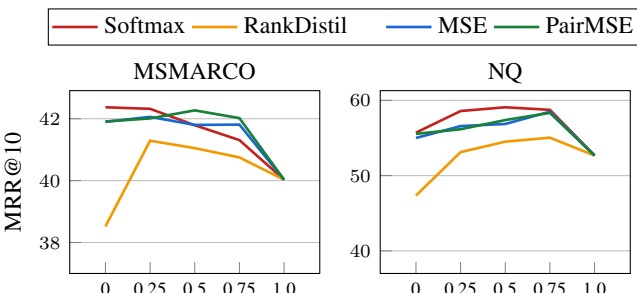

Figure 1: The ranking distillation performance on MSMARCO and NQ (T1) for different methods after varying the loss weights $\alpha$.

## 2.2 Performance with Softmax transformation on and off

In section The Role of Softmax Transformation, we briefly discussed whether Softmax teacher score transformation was selected in the parameter sweep for each method. Here we provide more details. The results on T1 are reported in Tbl. 2. We do not include RD and PairLog since they are indifferent to transformations. We do not include RankDistil either because Softmax transformation is always on by design. As we discussed in main text, different methods show different behaviors, and the difference made by turning teacher score transformation on and off can be huge.

## 2.3 Discussion on Large Language Model Rankers

Recently there is a strong interest in ranking models using Large Language Models (LLMs) [6]. This line of research is highly relevant to ranking distillation as the giant LLMs rankers usually need to

Table 2: Ranking distillation results on the MSMARCO and NQ datasets, measured by MRR@10, with Softmax teacher score transformation on and off. Rank means the actual rank (if this configuration is actually selected) or the hypothetical rank (if this configuration *were* selected) of the method in Table 2. Higher numbers (those were selected) are bolded.

| Models | MSMARCO | | | | NQ | | | |
| --- | --- | --- | --- | --- | --- | --- | --- | --- |
| | On | | Off | | On | | Off | |
| | MRR@10 | Rank | MRR@10 | Rank | MRR@10 | Rank | MRR@10 | Rank |
| MSE | 40.03 | 8 | **42.06** | 4 | 54.87 | 8 | **58.49** | 3 |
| PairMSE | 40.03 | 8 | **42.27** | 3 | 55.61 | 7 | **58.34** | 5 |
| GumbelNDCG | **41.85** | 6 | 40.30 | 7 | **57.90** | 6 | 53.06 | 8 |
| Softmax | **42.37** | 1 | 40.03 | 8 | **59.08** | 1 | 52.67 | 8 |
| LambdaLoss | **42.30** | 2 | 41.87 | 6 | **58.36** | 4 | 56.52 | 7 |

54  be distilled to a servable ranker. One future direction is to extend RD-Suite to consider LLM based
55  ranking teachers. We believe there are several unique research directions on ranking distillation meets
56  LLMs. For example, given the strong performance and generality of LLM rankers and the rapid
57  development of powerful LLMs, it could be a good timing to revisit the false negative issue on certain
58  datasets (e.g., MSMARCO).