# OpenReview forum: "RD-Suite: A Benchmark for Ranking Distillation"
_NeurIPS.cc/2023/Track/Datasets_and_Benchmarks — NeurIPS 2023 Datasets and Benchmarks Poster_

### Official Review · Reviewer_KNat · 2023-07-20
**Review for RD-Suite:  A Benchmark for Ranking Distillation**

**Rating:** 5
**Confidence:** 3
**Clarity:** The paper is thoughtfully written and…

**Strengths:**

-It includes two modalities and formulates a novel distillation transfer task.
-The paper is well-written and easy to follow.
-The source codes and dataset have be released.

**Additional Feedback:**

N/A

**Correctness:**

This submission is a benchmark, and the experiment design and evaluation are performed correctly.

**Documentation:**

This codebase lacks detailed documentation now

**Limitations:**

No, the authors do not discuss the limitations of their work. Please see Opportunities For Improvement.

**Opportunities For Improvement:**

-The authors raise three questions, but do not take these three questions into consideration in the design of the benchmark. It would be more instructive if they could delve into Q1 and Q2.
-The experimental results are not well explained. Why are RD and RankDistl that are designed for LTR not even as effective as the simplest MSE? Does it mean that there is no need to design a distillation strategy specifically for LTR?
-In the codebase, there are only evaluation scripts, no LTR distillation framework, related models, and loss functions.

**Relation To Prior Work:**

Yes

**Summary And Contributions:**

This paper proposes a benchmark for ranking distillation. It includes 4 popular public datasets and formulates 2 applications (standard distillation and distillation transfer). The authors conduct experiments on several ranking losses.

---

> ### Author Response · Authors · 2023-08-20
> **Thank you very much for your feedback**
>
> Thank you very much for your feedback.
>
> 1. A main point of the benchmark is to raise awareness in this important field. Existing methods such as RD and RankDistil were developed with strict constraints on teacher models and only considers ordering as teacher labels. In Q1, we draw an analogy to knowledge distillation in classification where logit values are used, which RD/RankDistil do not consider. In Q2, we provide more discussion on the complexity of real-world teacher rankers. We believe RD-Suite will pave ways to future research to develop novel and more solid distillation specific methods (though it is not the focus of this benchmark paper).
>
> 2. Yes, so far we released data and evaluation scripts. We plan to release more code and keep updating RD-Suite in the future. Right now, given that the student architectures are well-known models, e.g., BERT and linear models, researchers can already start building novel approaches on our benchmark datasets. In addition, we have a discussion about model implementation in Appendix L53.

---

### Official Review · Reviewer_1Yku · 2023-07-20
**This paper proposes a systematic and unified benchmark that is valuable to the community.**

**Rating:** 8
**Confidence:** 3
**Correctness:** Yes
**Clarity:** Yes

**Strengths:**

1. This paper is well-written;
2. Reproducible code is provided;
3. This paper proposes a new benchmark ranking distillation that fills the gap of lack of consensus in the field.


**Additional Feedback:**

NA

**Documentation:**

Yes

**Limitations:**

1. I am not familiar with the field, what is dark knowledge in line 85?


**Opportunities For Improvement:**

It is expected that the authors will publish more datasets and evaluation scripts to encourage future research and fair comparisons in the field.

**Relation To Prior Work:**

Yes

**Summary And Contributions:**

The paragraphs discuss the intersection of learning to rank (LTR) and knowledge distillation in the context of neural ranking models. It highlights the lack of consensus on suitable benchmarking test beds for ranking distillation methods and the diversity in task types, datasets, and ranker configurations, making fair comparisons challenging.

To address this, the authors propose a new benchmark called Ranking Distillation Suite (RD-Suite), which includes sensible teacher and student model architectures, standard distillation, and a novel distillation transfer setting. They plan to release datasets and evaluation scripts to encourage future research and fair comparisons in this field.

The benchmark not only focuses on empirical comparisons but also aims to understand open questions and limitations of existing methods in ranking distillation. The authors raise three questions related to methodology and reproducibility, challenging common wisdom in the field. They provide discussions to shed light on empirical results, rationalize design choices, and present research challenges for the community to tackle.

In summary, the contribution of this work includes a comprehensive ranking distillation benchmark, analysis of key issues challenging prior art, and the release of accessible datasets and evaluation scripts to support future research in the area.

---

> ### Author Response · Authors · 2023-08-20
> **Thank you very much for your feedback**
>
> Thank you very much for your feedback.
>
> Dark knowledge is a term that usually refers to what the student model distill from, proposed by Hinton (https://www.ttic.edu/dl/dark14.pdf) and even used in some papers (https://arxiv.org/pdf/2107.02629.pdf). We agree it may not be a very scientific term. Since this is also raised by another reviewer, we will just use “knowledge” in the revision as in the original knowledge distillation paper (https://arxiv.org/pdf/1503.02531.pdf).

---

### Official Review · Reviewer_rCz5 · 2023-07-21

**Rating:** 7
**Confidence:** 3
**Correctness:** To the best of my knowledge, the prov…
**Clarity:** Yes, the paper and reproduction scrip…

**Strengths:**

1. The four datasets cover a wide range of tasks (including text ranking and tabular ranking), with varying degrees of transferability, making the benchmarking comprehensive.
2. The GitHub repository is well-established. The reproduction scripts provided by the authors are clear.
3. The authors discuss several open questions based on the benchmarking results.
4. The paper is well-written and easy-to-follow.

**Additional Feedback:**

N/A

**Documentation:**

Yes, the scripts to reproduce the results are clear and easy-to-follow.

**Opportunities For Improvement:**

1. Although this paper provides a range for hyperparameter tuning (Table 2 in the supplementary material) to assist readers in reproducing the results, authors are encouraged to further publicize the optimal hyperparameters corresponding to each result in the benchmarks.
2. Line 47 of the supplementary material has rendering issue, i.e. Sec ??.

**Relation To Prior Work:**

The discussion on existing works are well conveyed in Section 7.

**Summary And Contributions:**

This paper sets benchmarks for four datasets in the direction of ranking distillation. These four datasets cover a wide range of tasks (including text ranking and tabular ranking), with varying degrees of transferability. The paper also discusses several open questions based on the benchmark results. The GitHub repository scripts provided by the authors are clear and easy to follow.

---

> ### Author Response · Authors · 2023-08-20
> **Thank you very much for your feedback**
>
> Thank you very much for your feedback.
>
> 1. We have the hyperparameters on record and will add a section in the appendix. Thank you for your feedback.
>
> 2. We really appreciate your careful review. It is referring to Section 5.5 in the main text. We will fix it in the revision.

---

### Official Review · Reviewer_uUSL · 2023-07-22
**First unified benchmark to evaluate ranking distillation**

**Rating:** 6
**Confidence:** 1
**Correctness:** See opportunities for improvement.
**Clarity:** See opportunities for improvement.

**Strengths:**

[significance of the contribution]. The paper tries to address a challenging problem of benchmarking the rank distillation loss function performance. The challenge stems from the varied application scenarios, vagueness in evaluating metrics etc. The paper makes some first systemic efforts to address these issues, aiming to provide a unified evaluation benchmarks.
[relevance to the broader research]. The rank distillation is an increasingly hot problem. The methodology provided in this paper may inspire a more thorough and rigorous designed benchmark.
[quality of the research]. The paper thoroughly examines existing methods and identifies, investigates and mitigates some critical issues in designing such a benchmark to make it more authoritative.

**Additional Feedback:**

I could not well evaluate this paper for two reasons:
1. I'm not familiar with ranking distillation field.
2. The English writing needs significant improvements for readers to understand this paper.

**Documentation:**

See opportunities for improvement.

**Ethics:**

No concern.

**Limitations:**

See opportunities for improvement.

**Opportunities For Improvement:**

The primary issue of this paper is the presentation.
1. The ranking distillation problem is not well introduced. After several passes, I found the problem is about how to design or choice the loss function for distillation. (Correct me if I misunderstood).
2. The three questions Q1-Q3 are not well presented. This could be improved by first giving some high level idea and giving concrete examples of the consequence beyond the math results if these questions are not handled. Meanwhile, the answers to these questions are not explicitly stated.
3. Many discussion, arguments and justifications are given without introducing some basic background. For example, the `dark knowledge`.
4. The English needs polished before publication. It's a bottleneck to understand this paper.

Results:
The paper reveals that some specifically designed loss function RD and RankDistil are not competitive to some more common loss function such as softmax. This is an interesting finding. However, the paper does not provide any insights or discussion the reason to this results.

**Relation To Prior Work:**

Yes, clearly discussed.

**Summary And Contributions:**

This paper provides a benchmark to evaluation the performance of different distillation loss function for ranking distillation tasks. To evaluate the loss functions, this benchmark uses a fixed set of teacher and student model and evaluates on four real-world datasets with four type of ranking tasks (textual, numeric) x (standard distillation, distillation transfer) with five ranking metrics. The paper also tries to address three critical issues in establishing such a dataset: 1. where is the knowledge loss of the distillation? 2. how to better handle the teacher ranker scores to remove the ambiguity related to ranking? 3. how to make sure the evaluation is reproducible and fair. The paper reveals the performance of different distillation loss functions on different tasks and metrics and shows that some specifically designed loss function RD and RankDistil are not competitive to some more common loss function such as softmax.

---

> ### Author Response · Authors · 2023-08-20
> **Thank you very much for your feedback.**
>
> Thank you very much for your feedback.
>
> 1. You are correct, this paper mainly focuses on the objective/loss of ranking distillation. Please see discussion in Section 4.1 D1. Most relevant references ([32][29]) also focus on the objective though there could be other architecture specific distillation methods that depend on the teacher architecture. We will try to make this more clear in the revision.
>
> 2. We will try to make the questions more clear in the revision.  We agree with you, the answers to the questions are scattered in the paper (e.g., L189). We plan to have a summary section to answer the questions.
>
> 3. We will try to go over the paper and make confusing terms more clear. “Dark knowledge” is commonly used when referring to knowledge distillation (https://www.ttic.edu/dl/dark14.pdf) and we agree it might not be a very scientific term. We will just use “knowledge” in the revision as in the original knowledge distillation paper (https://arxiv.org/pdf/1503.02531.pdf).
>
> 4. We will try our best to polish the paper in the revision, incorporating the feedback above.

---

### Decision · Program_Chairs · 2023-09-22

**Decision:**

Accept (Poster)

**Comment:**

This paper tries to address the challenge of benchmarking the rank distillation loss function performance. A benchmark is proposed to evaluate different distillation loss function for the ranking distillation tasks. All reviewers agree that the paper makes systemic efforts, and the proposed benchmarking is comprehensive. The released datasets and evaluation scripts can support the future research in the area. The discussion aims to understand open questions and limitations of existing methods in ranking distillation. The reviewers also have concerns in that the raised three questions are not explicitly taken into consideration during the design of the benchmark. In the discussion phase, the authors answered most of the review questions.